# FR-GAN: FAIR AND ROBUST TRAINING

## ABSTRACT

We consider the problem of *fair* and *robust* model training in the presence of data poisoning. Ensuring fairness usually involves a tradeoff against accuracy, so if the data poisoning is mistakenly viewed as additional bias to be fixed, the accuracy will be sacrificed even more. We demonstrate that this phenomenon indeed holds for state-of-the-art model fairness techniques. We then propose FR-GAN, which *holistically performs fair and robust model training* using generative adversarial networks (GANs). We first use a generator that attempts to classify examples as accurately as possible. In addition, we deploy two discriminators: (1) a fairness discriminator that predicts the sensitive attribute from classification results and (2) a robustness discriminator that distinguishes examples and predictions from a clean validation set. Our framework respects all the prominent fairness measures: disparate impact, equalized odds, and equal opportunity. Also, FR-GAN optimizes fairness without requiring the knowledge of prior statistics of the sensitive attributes. In our experiments, FR-GAN shows almost no decrease in fairness and accuracy in the presence of data poisoning unlike other state-of-the-art fairness methods, which are vulnerable. In addition, FR-GAN can be adjusted using parameters to maintain reasonable accuracy and fairness even if the validation set is too small or unavailable.

## 1 INTRODUCTION

As machine learning becomes widespread in the Software 2.0 era (Karpathy, 2017), there is an urgent need to address the issues of fairness and robustness. For sensitive applications like healthcare, finance, and self-driving cars, a trained model must not discriminate customers based on sensitive attributes including age, sex, or religion. In addition, as applications often rely on external data sets for their training data, the model training must be resilient against noisy or even adversarial data.

Traditionally, model fairness research (Venkatasubramanian, 2019; Chouldechova & Roth, 2018; Verma & Rubin, 2018) has focused on developing metrics such as disparate impact (Feldman et al., 2015), equalized odds (Hardt et al., 2016), and equal opportunity (Hardt et al., 2016), which capture various notions of discrimination. More recently, there has been a surge in *unfairness mitigation* techniques (Bellamy et al., 2018), which improve the model fairness by either fixing the training data, training process, or trained model. Unfairness mitigation usually involves some tradeoff between the model's accuracy and fairness. Most recently, generative adversarial networks (GANs) are being adapted to a fairness setting (Zhang et al., 2018). The architecture of GANs is suitable because it can be used to compete the training between two objectives: accuracy and fairness.

Robust model training is also important and needs to be concurrently taken into consideration. There has been a proliferation of algorithms that make model training resilient against noisy and even adversarial data (Natarajan et al., 2013; Biggio et al., 2011; Frénay & Verleysen, 2014). The latter scenario may occur when the source of the training data is external, and we cannot control the data that is being added. For example, any public dataset on the Web may contain intentionally-false information. Moreover, one must have a robust training algorithm if the data set comes from diverse users' inputs that cannot be fully trusted (Konečný et al., 2016). Recently, data poisoning attacks have become increasingly sophisticated where it is becoming difficult to sanitize the data against all of them (Koh et al., 2018). In addition, dataset searching is becoming mainstream as demonstrated by Google Dataset Search (Goods) (Halevy et al., 2016) and its public version for searching scientific datasets (Noy et al., 2019). While data lakes within companies may consist of refined datasets, datasets in the public are easy to poison. And anyone can poison public datasets

using attacks in the literature and share them. We thus believe that it is essential to address both bias and poisoning as a preventive measure.

Solving model fairness without addressing data poisoning may lead to a worse tradeoff between accuracy and fairness. For example, consider a banking system that is giving out loans where there are two sensitive groups: men and women. According to disparate impact, the two groups must have similar ratios of positive predictions. A perfect disparate impact would have a value of 1. Suppose the clean training data has no bias and a perfect (or high) disparate impact score. Now suppose this data is poisoned where half of the positive labels of men are flipped to negative labels. If the model is fitted so as to make more negative predictions on men than women, then disparate impact would worsen. To address this problem, one may suggest using data sanitization techniques *prior to* the model training, but it is challenging to do so without any knowledge of the model. Even though many data sanitization methods are proposed, there have been newer attacks that can easily evade such defenses (Koh et al., 2018).

Our main contribution is the development of a new framework called FR-GAN, which trains accurate models that are also fair and robust to poisoning. FR-GAN consists of a generator used for classification, a discriminator that distinguishes predictions from one sensitive group against others, and a second discriminator that distinguishes {examples, predictions} of the training data from {examples, labels} of a separate and clean validation set. The first discriminator ensures that the prediction $\hat{y}$ is independent of the sensitive attribute $z$. We show that variations of this approach can be used to maximize any of the following prominent fairness measures: disparate impact, equalized odds, and equal opportunity. The second discriminator ensures that the model predictions on the training data are "consistent" with labels on clean data. We theoretically show that FR-GAN optimizes fairness without requiring the knowledge of prior statistics of sensitive attributes. In addition, the parameters of FR-GAN can be adjusted to maintain reasonable accuracy and fairness even if the validation set is too small or unavailable. In the following sections, we present the related work, demonstrate the weaknesses of current fairness methods, and propose FR-GAN.

## 2 RELATED WORK

**Model fairness**    The notion of discrimination has many definitions and usually comes from certain social goals that one wants to guarantee. Fairness research has been focused on defining fairness measures that capture a variety of notions of fairness (Verma & Rubin, 2018). There are largely two types of fairness measures: (1) group fairness (Barocas & Selbst, 2016; Hardt et al., 2016), which ensures similar statistics between two sensitive groups; (2) individual fairness (Dwork et al., 2012), which guarantees similar prediction results across nearby examples. Group fairness measures are widely studied in general applications, even though these cannot be simultaneously satisfied in certain circumstances (Kleinberg et al., 2017). In addition, there are also recent significant works that focus on individual fairness (Garg et al., 2018; Yurochkin et al., 2019; Kearns et al., 2019; Jung et al., 2019). Among the two types of fairness with different aspects, we focus on group fairness as our initial research. Recently, there has also been a surge of research on unfairness mitigation techniques (Bellamy et al., 2018). Depending on where a fix occurs, there are mainly three approaches: (1) *pre*-processing techniques (Kamiran & Calders, 2011; du Pin Calmon et al., 2017; Zemel et al., 2013; Feldman et al., 2015) that fix the training data; (2) *in*-processing techniques (Zafar et al., 2017; Jiang & Nachum, 2019; Zhang et al., 2018; Kamishima et al., 2012; Cotter et al., 2019; 2018; Agarwal et al., 2018) that address the issue during model training; and (3) *post*-processing techniques (Hardt et al., 2016; Pleiss et al., 2017; Kamiran et al., 2012) that manipulate predictions while maintaining the model.

Among the three, the in-processing techniques have the advantages that one can work with any data and that there is more control on model training (Venkatasubramanian, 2019). FAIRNESS CONSTRAINTS (Zafar et al., 2017) incorporates a regularization term that reflects fairness constraints in the context of convex margin-based classifiers such as logistic regression and support vector machines (SVMs). The main hyperparameter is the $\lambda$ term, which serves to balance the accuracy and fairness objectives. LABEL BIAS CORRECTION (Jiang & Nachum, 2019) assumes the existence of true yet *possibly biased* labels. A reweighting algorithm is proposed with theoretical guarantees that training on the resulting loss corresponds to training on the true *unbiased* labels, which yields a fair model. Here the hyperparameters include the number of training iterations. Finally, ADVERSARIAL

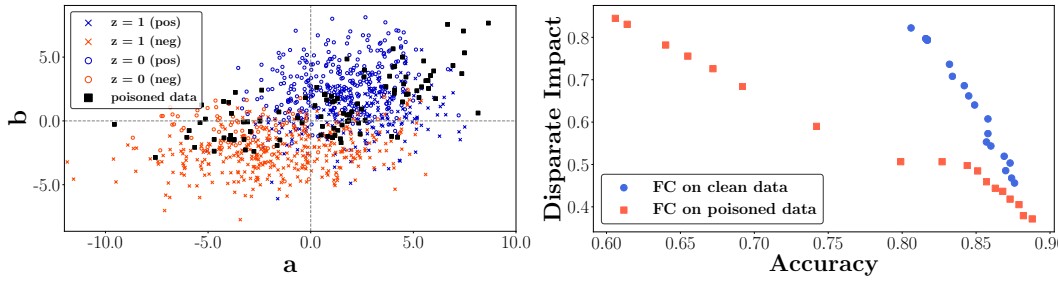

(a) Synthetic data with z-flipped poisoning

(b) Accuracy-fairness tradeoff for FAIRNESS CONSTRAINTS (Zafar et al., 2017)

Figure 1: The left figure shows a synthetic dataset with a certain poisoning. Examples are divided into $z = 0$ (marked with circles) and $z = 1$ (crosses) as per a sensitive attribute $z$. The blue points indicate positive labels while the red points denote negative ones. For data poisoning, we flipped $z$ values that reduce the model accuracy the most (similar to label flipping (Paudice et al., 2018), but specialized for fairness). The right figure shows that poisoning significantly worsens the accuracy-fairness tradeoff (i.e., shifts to the left) of the FAIRNESS CONSTRAINTS method (Zafar et al., 2017).

DEBIASING (Zhang et al., 2018) trains a classifier so that an adversary cannot predict the sensitive attributes based on the output of the classifier. Hence, the classifier achieves fairness when its predictions do not have any correlation with sensitive attributes. This idea of adversarial training is similar to GANs (Goodfellow et al., 2014). ADVERSARIAL DEBIASING provides insights of using adversarial training to minimize mutual information between the prediction and sensitive attributes. In this paper, we strengthen the theoretical results of ADVERSARIAL DEBIASING using information theory, and this motivates us to provide systematic methodology for various fairness metrics (Section 4.1).

As we demonstrate in Section 3, the existing techniques are not tailored for robust training, so are vulnerable to data poisoning attacks. In comparison, FR-GAN addresses both model fairness and robust training within the same model training process.

**Robust training** There is a heavy literature on how to make the model training robust against noisy or even adversarial data (Natarajan et al., 2013; Biggio et al., 2011; Frénay & Verleysen, 2014; Kurakin et al., 2017). A major challenge is that there can be a wide range of data poisoning attacks that keep on evolving. One defense approach is to sanitize the training data and thus fix the root cause. A fundamental limitation is that we do not know in general what kind of poisoning will occur, so a universal defense against all possible attacks does not seem feasible as demonstrated by Koh et al. (2018). A more recent trend is to develop general defense algorithms for any attack *during model training* using meta learning (Veit et al., 2017; Li et al., 2017; Xiao et al., 2015; Hendrycks et al., 2018). Ren et al. (2018) propose a meta-learning framework that employs a clean validation set for the purpose of preventing the model training from being influenced by poisoned data. By using the validation loss as a meta objective, it intends to defend against any poisoning attack. Our FR-GAN framework is inspired by this approach, but employs a GAN-based model to support fair and robust training without using meta learning. In particular, the design of FR-GAN's robustness discriminator is based on mutual-information-based theoretical insights (Section 4.2). Another line of research is defending against adversarial attacks during *test* time (Biggio et al., 2013; Goodfellow et al., 2015; Wong & Kolter, 2018). In comparison, our focus is on defending against data poisoning on the *training* data.

## 3 VULNERABILITY OF FAIRNESS METHODS

We perform experiments to demonstrate that state-of-the-art fairness methods are indeed vulnerable even to simple poisoning attacks. We generate a synthetic dataset as shown in Figure 1a (see the generation details in Section 5.1). There are two non-sensitive attributes $a$ and $b$, which are reflected in the x-axis and y-axis, respectively. The examples are further divided into two classes based on the *sensitive* attribute $z$. To generate poisoned data, we measure accuracy degradation for a flipping of each $z$. We then choose the top-10% of the $z$ values with the highest degradation and flip them. This approach is similar to the existing label flipping method (Paudice et al., 2018), except that the

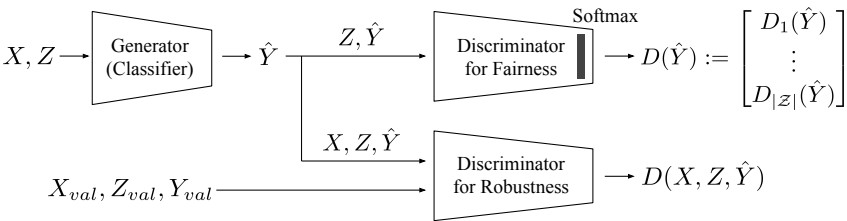

Figure 2: The architecture of FR-GAN.

$z$ value is being flipped instead of the label so that the model fairness is also affected. To make a validation set, we randomly select clean examples that amount to 10% of the entire training data.

We use disparate impact (Feldman et al., 2015) as a fairness measure and evaluate one fairness method, called FAIRNESS CONSTRAINTS (Zafar et al., 2017). As this method involves a regularization factor $\lambda$ that balances the accuracy and fairness objectives, we can obtain a tradeoff curve by adjusting its value. Figure 1b shows two accuracy-fairness tradeoff curves obtained with the clean and poisoned synthetic datasets. Notice that adding data poisoning clearly shifts the curve to the left, which means both accuracy and fairness decrease. This coincides with our intuition. The poisoning confuses the model so that there are more biased examples to fix, which in turn makes it *overreact* and thus sacrifice more on accuracy. In the supplementary, we show similar trends when the data is poisoned via label flipping. In Section 5, we will show how data poisoning affects LABEL BIAS CORRECTION (Jiang & Nachum, 2019) and ADVERSARIAL DEBIASING (Zhang et al., 2018).

## 4   FR-GAN

We now describe FR-GAN, which is shown in Figure 2. Unlike traditional GANs, the generator is a classifier that receives an example $x \in X$ and returns a prediction $\hat{y}$. The other two discriminators optimize fairness and robustness as we describe in the following sections.

### 4.1   FAIRNESS

We denote by $\mathcal{D}_{tr}$ the training data set. Suppose $\mathcal{D}_{tr}$ has $m$ examples $\{(x^{(i)}, z^{(i)}, y^{(i)})\}_{i=1}^{m}$ where $x^{(i)}$ contains the non-sensitive attributes, $z^{(i)}$ contains the sensitive attributes, and $y^{(i)}$ is the label. Both the sensitive attribute and label can be multi-class, i.e., they can have one of multiple values. For notational simplicity, we assume there is one sensitive attribute, which can be viewed as a merged result of multiple sensitive attributes with a larger alphabet size. For illustrative purposes, we focus on disparate impact, leaving in the supplementary our formulation for equalized odds and equal opportunity.

The first discriminator in FR-GAN distinguishes predictions w.r.t. one sensitive group from those in the others. Disparate impact intends the sensitive attribute to be independent of the model's prediction, i.e.,

$$I(Z; \hat{Y}) = 0.$$

We explain how FR-GAN can enforce the above constraint. Let $P_Z(z)$ be the distribution of $Z$ where $z \in \mathcal{Z}$ and $\mathcal{Z}$ is the set of possible sensitive attribute values. Let $\hat{Y}|Z = z \sim P_{\hat{Y}|z}(\cdot)$ and $\hat{Y} \sim P_{\hat{Y}}(\cdot)$. Then $P_{\hat{Y}}(\cdot) = \sum_{z \in \mathcal{Z}} P_Z(z) P_{\hat{Y}|z}(\cdot)$.

We show that mutual information is equivalent to the following expression where the optimal discriminator $D_z^\star(\hat{y}) = \frac{P_{\hat{Y},z}(\hat{y},z)}{P_{\hat{Y}}(\hat{y})}$ and $\sum_{z \in \mathcal{Z}} D_z^\star(\hat{y}) = 1 \ \forall \hat{y} \in \mathcal{Y}$.

**Theorem 1.** $I(Z; \hat{Y}) = \max_{D_z(\hat{y}): \sum_z D_z(\hat{y})=1 \forall \hat{y}} \sum_{z \in \mathcal{Z}} P_Z(z) \mathbb{E}_{P_{\hat{Y}|z}} \left[ \log D_z(\hat{Y}) \right] + H(Z).$

While deferring the detailed proof to the supplemental materials, we provide a brief overview of the proof. As the optimization problem in the RHS is convex, we find the optimal dicriminator by solving the KKT conditions. We then show that the maximum value attained by the optimal discriminator is equal to the mutual information by using the properties of mutual information and the generalized Jensen-Shannon divergence (Lin, 1991).

What is more involved than showing the above equality is designing a right optimization problem. One needs to carefully handcraft a plausible optimization problem so that its unique solution matches the desired quantity. Here, we design the above optimization problem via a 'guess-&-check' approach aided by the structural insights across the KL divergences that appear in an alternative expression of mutual information.

We now discuss how to implement the above expression. Since we do not know $P_{\hat{Y}|z}(\cdot)$ exactly, we compute the following empirical version:

$$\max_{D_z(\hat{y}):\sum_z D_z(\hat{y})=1\forall\hat{y}} \sum_{z\in\mathcal{Z}} P_Z(z) \sum_{i:z^{(i)}=z} \frac{1}{m_z} \log D_z(\hat{y}^{(i)}) + H(Z).$$

Now for sufficiently large $m$, the number $m_z$ of examples with $z^{(i)} = z$ is approximately the same as $P_Z(z)m$. Therefore, the above expression becomes:

$$\max_{D_z(\hat{y}):\sum_z D_z(\hat{y})=1\forall\hat{y}} \sum_{z\in\mathcal{Z}} \sum_{i:z^{(i)}=z} \frac{1}{m} \log D_z(\hat{y}^{(i)}) + H(Z).$$

Interestingly, this formulation is exactly the same as that in the original GAN (Goodfellow et al., 2014) when $|\mathcal{Z}| = 2$. We also remark that our formulation does not require a prior knowledge on $P_Z(z)$.

## 4.2 ROBUSTNESS

We utilize a clean yet small validation set to ensure robust training. In particular, we add a second discriminator that distinguishes the training data with predictions $\{(x^{(i)}, z^{(i)}, \hat{y}^{(i)})\}_{i=1}^m$ from the validation set $\{(x_{\text{val}}^{(i)}, z_{\text{val}}^{(i)}, y_{\text{val}}^{(i)})\}_{i=1}^{m_{\text{val}}}$. Intuitively, if the classifier is confused by data poisoning in the training data, then its predictions will not be consistent with the labels of the clean data, and the discriminator would be able to detect that difference. While using a validation set is inspired by the meta learning approach by Ren et al. (2018), we take an adversarial learning approach. A key motivation behind this choice is that our GAN approach introduces a knob that controls the emphasis of robust training. We find that this knob enables FR-GAN to be more robust to the validation set size. See details in Section 5.1.

Our formulation is as follows. Let $V \sim Bern(\beta)$ denote whether an example comes from training data ($V = 1$) or from the validation set ($V = 0$). We then want to ensure that $V$ is independent of the training data and predictions, i.e., $I(V; X, Z, \hat{Y}) = 0$, which means the predictions on the training data are indistinguishable from labels of the validation set. This way we can mimic the clean data set while expecting an indirect sanitization effect. We optimize this independence using a formulation similar to that of fairness. Another possible formulation is to use the hard decision value from $D(\hat{Y})$, say $\hat{Z}$, instead of $Z$. However, this discriminator can only be trained *after* the fairness discriminator is trained. Instead, we take an approach that allows us to train the two discriminators at the same time.

## 4.3 ARCHITECTURE

Figure 2 illustrates the architecture of FR-GAN. For the loss function in the generator, we employ the cross entropy loss:

$$L_1 = \frac{1}{m} \sum_{i=1}^m -y^{(i)} \log \hat{y}^{(i)} - (1 - y^{(i)}) \log(1 - \hat{y}^{(i)}).$$

As per Theorem 1, we set the loss function w.r.t. the fairness discriminator as:

$$L_2 = \max_{D(\cdot)} \sum_{z\in\mathcal{Z}} \sum_{i:z^{(i)}=z} \frac{1}{m} \log D_z(\hat{y}^{(i)}) + H(Z)$$

where $D(\cdot) := (D_1(\cdot), \ldots, D_{|\mathcal{Z}|}(\cdot))$. The condition $\sum_{z\in\mathcal{Z}} D_z^\star(\hat{Y}) = 1$ can be enforced by adding a softmax layer to the discriminator.

Table 1: Accuracy and fairness performances on the clean and poisoned synthetic datasets w.r.t. disparate impact. Two types of methods are compared: (1) non-fairness methods: logistic regression (LR), the meta learning by Ren et al. (2018) (ML), and R-GAN (FR-GAN with $\lambda_1 = 0$); (2) fairness-only methods: FAIRNESS CONSTRAINTS (Zafar et al., 2017) (FC), LABEL BIAS CORRECTION (Jiang & Nachum, 2019) (LBC), and ADVERSARIAL DEBIASING (Zhang et al., 2018) (AD). Also "+ LD" denotes loss defense (Koh et al., 2018) (a data sanitization technique that is applied prior to model training). For FR-GAN, the validation set is 10% of $\mathcal{D}_{tr}$. For each fairness or accuracy result of the poisoned data, we make a comparison with the clean data result and show the percentage increase or decrease.

| Type | Method | Clean data | | Poisoned data | |
|------|--------|-----------|------|--------------|------|
| | | D. impact | Acc. | D. impact | Acc. |
| Non-fair | LR | 0.409 | 0.885 | 0.436 (6.60% ↑) | 0.876 (1.02% ↓) |
| | ML | 0.429 | 0.883 | 0.386 (10.0% ↓) | 0.884 (0.11% ↑) |
| | R-GAN | 0.414 | 0.883 | 0.338 (18.4% ↓) | 0.884 (0.11% ↑) |
| Fair-only | FC | 0.822 | 0.806 | 0.756 (8.03% ↓) | 0.655 (18.7% ↓) |
| | LBC | 0.819 | 0.760 | 0.816 (0.37% ↓) | 0.738 (2.89% ↓) |
| | AD | 0.807 | 0.811 | 0.711 (11.9% ↓) | 0.812 (0.12% ↑) |
| | FC+LD | 0.822 | 0.806 | 0.763 (7.18% ↓) | 0.651 (19.2% ↓) |
| | LBC+LD | 0.819 | 0.760 | 0.513 (37.4% ↓) | 0.859 (13.0% ↑) |
| | AD+LD | 0.807 | 0.811 | 0.730 (9.54% ↓) | 0.818 (0.86% ↑) |
| Both | FR-GAN | 0.813 | 0.809 | 0.830 (2.09% ↑) | 0.808 (0.12% ↓) |

Finally, implementing $I(V; X, Z, \hat{Y})$ as advised by Theorem 1, we set the loss function w.r.t. the robustness discriminator as:

$$L_3 = \max_{D^r(\cdot)} \sum_{i:v^{(i)}=0} \frac{1}{m} \log D^r(x^{(i)}, z^{(i)}, \hat{y}^{(i)}) + \sum_{i:v^{(i)}=1} \frac{1}{m} \log(1 - D^r(x^{(i)}, z^{(i)}, \hat{y}^{(i)})) + H(V).$$

The final objective function is to minimize the weighted sum of these value functions:

$$\min_{G(\cdot)} L_1 + \lambda_1 L_2 + \lambda_2 L_3.$$

Here $\lambda_1$ and $\lambda_2$ are tuning knobs that play roles to emphasize fair and robust training, respectively.

## 5 EXPERIMENTS

We provide experimental results for disparate impact, while leaving in the supplementary those for equalized odds and equal opportunity. All of our results are on a separate test set. More implementation details are also in the supplementary.

### 5.1 SYNTHETIC DATA RESULTS

For the synthetic data, we generate 2,000 examples with two non-sensitive attributes $a$ and $b$, a sensitive attribute $z$, and a label $y$, using a method similar to the algorithm proposed by Zafar et al. (2017). Both $z$ and $y$ are binary and generated uniformly at random. For the four possible $(z, y)$ combinations, we generate different normal distributions for $a$ and $b$. Finally for each example, the $a$ and $b$ values are sampled as per the normal distribution associated with the $(z, y)$ pair. For data poisoning, we flip the $z$ values of examples so as to lower the model's accuracy the most as described in Section 3.

To make a fair comparison in the context of robust training, we also improve the existing fairness methods with data sanitization and compare them with FR-GAN. Since FR-GAN is one of the first works to address both fairness and robustness, there are few baselines to compare with. Hence, we employed one reasonable baseline, which first sanitizes the poisoned data using a well-known sanitization technique and then performs each fairness algorithm. There are several sanitization techniques surveyed in Koh et al. (2018), and we choose the *loss defense* method. Here examples that

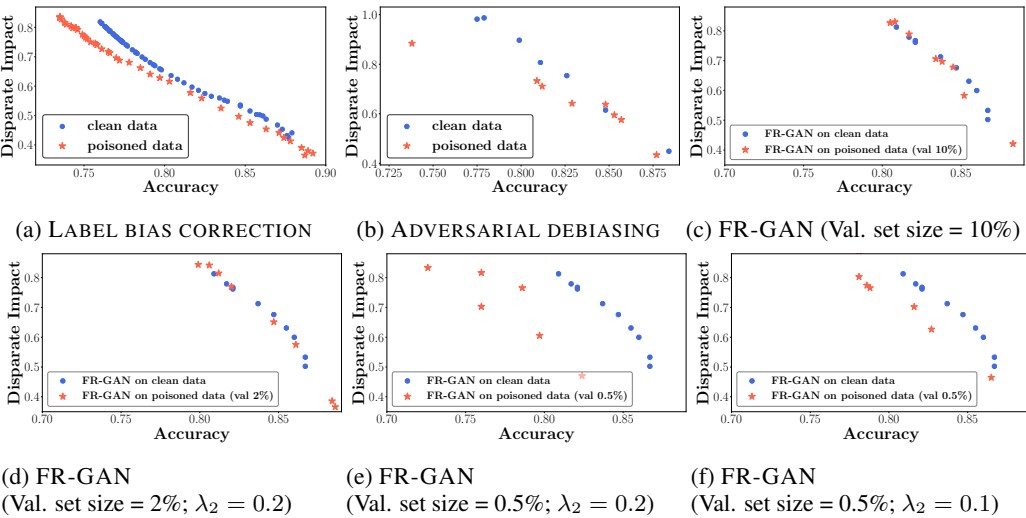

(a) LABEL BIAS CORRECTION  (b) ADVERSARIAL DEBIASING  (c) FR-GAN (Val. set size = 10%)

(d) FR-GAN
(Val. set size = 2%; $\lambda_2 = 0.2$)

(e) FR-GAN
(Val. set size = 0.5%; $\lambda_2 = 0.2$)

(f) FR-GAN
(Val. set size = 0.5%; $\lambda_2 = 0.1$)

Figure 3: Accuracy-fairness tradeoff curves.

are not well fit by the model and result in a high loss are considered poisoned and are subsequently discarded. Loss defense is the only method that utilizes the trained model and thus has the best chance of defending poisoning attacks.

**Accuracy and fairness**  We compare FR-GAN with FAIRNESS CONSTRAINTS (Zafar et al., 2017), LABEL BIAS CORRECTION (Jiang & Nachum, 2019), ADVERSARIAL DEBIASING (Zhang et al., 2018), and a plain non-fairness method, logistic regression. We also compare with the meta learning approach by Ren et al. (2018), which focuses on robust training. For this comparison, we tailor our framework to a setting where $\lambda_1 = 0$ (no fairness constraint), and name it R-GAN. For FR-GAN, we use a validation set that amounts to 10% of $\mathcal{D}_{tr}$. When setting $\lambda_1$ and $\lambda_2$, we usually fix $\lambda_2$ to some value and then adjust $\lambda_1$. For the clean data, we apply proper hyperparameters so that the disparate impacts are similar (around 0.8) across all distinct methods. We do the same for the poisoned data. There is no such hyperparameter tuning for logistic regression and the meta learning by Ren et al. (2018), as the frameworks have no knob for adjusting fairness. We find that this leads to poor disparate impacts. For the three fairness methods, data poisoning aggravates performances in accuracy, fairness or both. For example, the disparate impact and accuracy of FAIRNESS CONSTRAINTS fall by 8.03% and 18.7%, respectively. On the other hand, the performance changes for FR-GAN are negligible: disparate impact increases by 2.09%, and accuracy decreases by 0.12%. Table 1 also shows that combining the fairness methods with loss defense (rows 6–8) does not always yield better accuracy and fairness. In fact, using the loss defense may lower the accuracy or fairness (e.g., LBC+LD has a disparate impact of 0.513 on poisoned data while LBC has 0.816). Perhaps this is because the data sanitization sometimes mistakenly discards clean data as well. The results demonstrate that even a data sanitization technique may not help much. In the supplementary, we also perform experiments using equalized odds and show that FR-GAN has similar benefits.

We observe how accuracy trades off with fairness on clean and poisoned datasets. The results for FAIRNESS CONSTRAINTS are shown in Figure 1b. For LABEL BIAS CORRECTION, we employ the number of training iterations as a knob to trade accuracy off fairness. As shown in Figure 3a, the tradeoff curve shifts to the left, which demonstrates a clear tradeoff degradation. For ADVERSARIAL DEBIASING, we employ the $\alpha$ parameter (Zhang et al., 2018) analogous to $\lambda_1$ as a knob to trade accuracy off fairness. We see in Figure 3b that the tradeoff curve again shifts to the left.

**Validation set size**  Figures 3c to 3e show how the validation set size affects the robustness of FR-GAN. In particular, we compare the accuracy-fairness tradeoff of FR-GAN on clean data and that on poisoned data while varying the size of the validation set. When running on poisoned data, we fixed $\lambda_2 = 0.2$ and varied $\lambda_1$. We see that even 2% validation set (Figure 3d) is sufficient to maintain the accuracy and fairness obtained on the clean data. When using 0.5% (Figure 3e), the validation set is too small and has an adverse effect on the training. However, our framework includes a tuning knob $\lambda_2$, so by decreasing $\lambda_2$ down to 0.1, we could de-emphasize robust training, thereby avoiding

the adverse effect (Figure 3f). This is in contrast to the meta learning approach, which suffers from a non-negligible performance degradation for a very small validation set. See details in the supplementary. The results show that *only a holistic* framework like FR-GAN *can achieve both* excellent model fairness and training robustness. In comparison, other methods tailored for only one of these objectives lose either accuracy, fairness or both.

**Ablation study** (Without 'R') For a small value of $\lambda_2$, we observe in Figure 3f that the accuracy-fairness tradeoff curve shifts to the left, just like other fairness-only methods. To make the comparison clearer, we also plot them together in the supplementary. (Without 'F') When $\lambda_1 = 0$, FR-GAN is expected to behave similarly to the other non-fair algorithms. The performance of FR-GAN with $\lambda_1 = 0$, which we dub as R-GAN, is reported in Table 1. One can observe that R-GAN achieves a poor fairness performance just like the other non-fair algorithms.

## 5.2 REAL DATA RESULTS

We use the following benchmark datasets: ProPublica COMPAS (Angwin et al., 2016) and Adult Census (Kohavi, 1996). For sensitive attributes, we use SEX for both datasets. We do not employ the German Credit data set (Dua & Graff, 2017) as its highly biased towards positive labels. With such dataset, it is nearly impossible to observe any meaningful tradeoff.

For data poisoning, we use the same method employed on synthetic data: flipping the $z$ values so as to maximize the accuracy performance degradation. While there may be stronger attacks, this poisoning attack is effective enough to fool the three fairness methods. In the supplementary, we also evaluate FR-GAN against label flipping attacks and show similar results. We use the same constructions for the validation set and poisoning as described in Section 3.

**Accuracy and fairness** See Table 2. As in Table 1, we apply proper hyperparameters so that disparate impacts are similar across all distinct methods, both for the clean and poisoned datasets. The results are similar to Table 1: logistic regression and meta learning exhibit poor disparate impacts; the three fairness methods have worse disparate impact and accuracy due to data poisoning; and FR-GAN again shows little degradation both in fairness and accuracy. Table 2 also shows that combining the fairness methods with loss defense (rows 6–8 and 15–17) does not always yield better accuracy and fairness and may even lower them, which is consistent to the results on synthetic data.

## 6 DISCUSSION

In this section, we discuss possible ways to extend FR-GAN.

**Subjective labels** FR-GAN can be generalized to a setting where labels are subjective instead of being clean or poisoned. Here the dataset without the undesirable human biases becomes the clean validation set. In general, given two datasets with different distributions where one of them is desirable, FR-GAN can train robustly against the other distribution.

**Constructing real datasets** We propose a general methodology for constructing a real dataset that has both bias and poisoning, as well as a "clean" validation set. Suppose that we are performing evaluation tasks (say loan decisions) where there is a high chance of human bias (i.e., poisoning) due to the high workload. We can construct a clean validation set by selecting a few tasks, assigning more evaluators, and taking a majority vote of the evaluations to minimize the bias. While employing more evaluators can be expensive, constructing a small validation set is sufficient for robust training, which makes this method practical.

**Generalizing to all attacks** An interesting question is whether FR-GAN can defend against all poisoning attacks. If we know all the possible attacks, we can construct a training set containing these attacks. In the more challenging case where we do not know which attacks even exist, there seems to be a fundamental limitation in protecting against the attacks. Generalization is a universal problem in machine learning where a model trained on one dataset is not guaranteed to perform well in another dataset with a different distribution. Although the generalization is a critical issue to address, it is currently beyond the scope of this paper.

Table 2: Accuracy and fairness performances on real data for disparate impact. Two types of methods are compared: (1) non-fairness methods: logistic regression (LR), the meta learning by Ren et al. (2018) (ML), and R-GAN (FR-GAN with $\lambda_1 = 0$); (2) fairness-only methods: FAIRNESS CONSTRAINTS (Zafar et al., 2017) (FC), LABEL BIAS CORRECTION (Jiang & Nachum, 2019) (LBC), and ADVERSARIAL DEBIASING (Zhang et al., 2018) (AD). Also "+ LD" denotes loss defense (Koh et al., 2018) (a data sanitization technique that is applied prior to model training). For FR-GAN, the validation set is 10% of $\mathcal{D}_{tr}$. For each fairness or accuracy result of the poisoned data, we compare with the clean data result and show the percentage increase or decrease.

| Dataset | Type | Method | Clean data | | Poisoned data | |
|---|---|---|---|---|---|---|
| | | | D. impact | Acc. | D. impact | Acc. |
| COMPAS | Non-fair | LR | 0.465 | 0.674 | 0.431 (7.31% ↓) | 0.667 (1.04% ↓) |
| | | ML | 0.493 | 0.680 | 0.164 (66.7% ↓) | 0.661 (2.79% ↓) |
| | | R-GAN | 0.574 | 0.683 | 0.393 (31.5% ↓) | 0.650 (4.83% ↓) |
| | Fair-only | FC | 0.777 | 0.682 | 0.790 (1.67% ↑) | 0.671 (1.61% ↓) |
| | | LBC | 0.866 | 0.671 | 0.807 (6.81% ↓) | 0.615 (8.35% ↓) |
| | | AD | 0.846 | 0.680 | 0.822 (2.84% ↓) | 0.655 (3.68% ↓) |
| | | FC + LD | 0.777 | 0.682 | 0.790 (1.67% ↑) | 0.668 (2.05% ↓) |
| | | LBC + LD | 0.866 | 0.671 | 0.822 (5.08% ↓) | 0.628 (6.41% ↓) |
| | | AD + LD | 0.846 | 0.680 | 0.836 (1.18% ↓) | 0.620 (8.82% ↓) |
| | Both | FR-GAN | 0.936 | 0.667 | 0.947 (1.18% ↑) | 0.660 (1.05% ↓) |
| Adult | Non-fair | LR | 0.328 | 0.847 | 0.230 (29.9% ↓) | 0.848 (0.12% ↑) |
| | | ML | 0.346 | 0.844 | 0.215 (37.9% ↓) | 0.843 (0.12% ↓) |
| | | R-GAN | 0.335 | 0.847 | 0.285 (14.9% ↓) | 0.834 (1.53% ↓) |
| | Fair-only | FC | 0.825 | 0.826 | 0.816 (1.09% ↓) | 0.828 (0.24% ↑) |
| | | LBC | 0.825 | 0.825 | 0.825 (0.00% -) | 0.795 (3.64% ↓) |
| | | AD | 0.850 | 0.767 | 0.679 (20.1% ↓) | 0.754 (1.69% ↓) |
| | | FC + LD | 0.825 | 0.826 | 0.798 (3.27% ↓) | 0.807 (2.30% ↓) |
| | | LBC + LD | 0.825 | 0.825 | 0.815 (1.21% ↓) | 0.811 (1.70% ↓) |
| | | AD + LD | 0.850 | 0.767 | 0.821 (3.41% ↓) | 0.728 (5.08% ↓) |
| | Both | FR-GAN | 0.870 | 0.819 | 0.866 (0.46% ↓) | 0.827 (0.98% ↑) |

## 7 CONCLUSION

We proposed FR-GAN, which is a holistic framework that enables both unfairness mitigation and robust training. Our key contribution is proposing a novel GAN architecture that enjoys the *synergistic effect* of combining two approaches: (1) employing a fairness discriminator that distinguishes predictions w.r.t. one sensitive group from others and (2) employing a robust discriminator that distinguishes training data with predictions from a clean yet small validation set. We showed that existing fairness methods are vulnerable to data poisoning, even when combined with data sanitization techniques. In comparison, FR-GAN is robust to the poisoning and can be adjusted to maintain reasonable accuracy and fairness even if the validation set is too small or unavailable. We showed that FR-GAN is guaranteed to maximize fairness and supports all the prominent fairness measures: disparate impact, equalized odds, and equal opportunity.

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

## A  APPENDIX

### A.1  PROOF FOR THEOREM 1

Before we present the proof of the main theorem, we first recall our notation. Let $P_Z(z)$ be the distribution of $Z$ where $z \in \mathcal{Z}$ and $\mathcal{Z}$ is the set of possible sensitive attribute values. Let $\hat{Y}|Z = z \sim P_{\hat{Y}|z}(\cdot)$ and $\hat{Y} \sim P_{\hat{Y}}(\cdot)$. Then $P_{\hat{Y}}(\cdot) = \sum_{z \in \mathcal{Z}} P_Z(z) P_{\hat{Y}|z}(\cdot)$. Also, let $Y \sim P_Y(\cdot)$.

For convenience, let us repeat the statement of Theorem 1 here:

$$I(Z;\hat{Y}) = \max_{D_z(\hat{y}):\sum_z D_z(\hat{y})=1, \forall \hat{y}} \sum_{z \in \mathcal{Z}} P_Z(z) \mathbb{E}_{P_{\hat{Y}|z}} \left[ \log D_z(\hat{Y}) \right] + H(Z).$$

We now prove the theorem.

*Proof.* Denote by $\boldsymbol{D}$ the collection of $D_z(\hat{y})$ for all possible values of $z$ and $\hat{y}$, and by $\boldsymbol{\nu}$ the collection of $\nu_{\hat{y}}$ for all values of $\hat{y}$. We can construct the Lagrangian function as follows:

$$\mathcal{L}(\boldsymbol{D}, \boldsymbol{\nu}) = \sum_{z \in \mathcal{Z}} P_Z(z) \mathbb{E}_{P_{\hat{Y}|z}} \left[ \log D_z(\hat{Y}) \right] + H(Z) + \sum_{\hat{y} \in \mathcal{Y}} \nu_{\hat{y}} \left( 1 - \sum_{z \in \mathcal{Z}} D_z(\hat{y}) \right).$$

We use the following KKT conditions:

$$\frac{\partial \mathcal{L}(\boldsymbol{D}, \boldsymbol{\nu})}{\partial D_z(\hat{y})} = P_Z(z) \frac{P_{\hat{Y}|z}(\hat{y})}{D_z^\star(\hat{y})} - \nu_{\hat{y}}^\star = 0, \qquad \forall (\hat{y}, z) \in \mathcal{Y} \times \mathcal{Z},$$

$$1 - \sum_{z \in \mathcal{Z}} D_z^\star(\hat{y}) = 0, \qquad \forall \hat{y} \in \mathcal{Y}.$$

Solving the two equations, we obtain $\nu_{\hat{y}}^\star = P_{\hat{Y}}(\hat{y})$ for all $\hat{y}$. Thus,

$$D_z^\star(\hat{y}) = \frac{P_Z(z) P_{\hat{Y}|z}(\hat{y})}{P_{\hat{Y}}(\hat{y})}.$$

Putting this to the above optimization,

$$\sum_{z \in \mathcal{Z}} P_Z(z) \mathbb{E}_{P_{\hat{Y}|z}} \left[ \log \frac{P_Z(z) P_{\hat{Y}|z}(\hat{Y})}{P_{\hat{Y}}(\hat{Y})} \right] + H(Z)$$

$$= \sum_{z \in \mathcal{Z}} P_Z(z) \mathbb{E}_{P_{\hat{Y}|z}} \left[ \log \frac{P_Z(z) P_{\hat{Y}|z}(\hat{Y})}{P_{\hat{Y}}(\hat{Y})} \right] + \sum_{z \in \mathcal{Z}} P_Z(z) \log \frac{1}{P_Z(z)}$$

$$= \sum_{z \in \mathcal{Z}} P_Z(z) \mathbb{E}_{P_{\hat{Y}|z}} \left[ \log \frac{P_{\hat{Y}|z}(\hat{Y})}{P_{\hat{Y}}(\hat{Y})} \right]$$

$$= \sum_{z \in \mathcal{Z}} P_Z(z) D_{\mathrm{KL}}(P_{\hat{Y}|z} \| P_{\hat{Y}})$$

$$\triangleq \mathrm{JS}_{P_Z}(P_{\hat{Y}|z_1}, \dots, P_{\hat{Y}|z_{|\mathcal{Z}|}}) = I(Z;\hat{Y}).$$

Here, the second last equality is due to the definition of the generalized JS divergence, and the last equality is due to its equivalence to the mutual information (Lin, 1991). □

### A.2  EXTENSIONS TO OTHER FAIRNESS MEASURES

The following theorem relates the conditional mutual information $I(Z;\hat{Y}|Y)$ to the solution of an optimization problem.

**Theorem 2.**

$$I(Z;\hat{Y}|Y) = \max_{D_{z|y}(\hat{y}):\sum_{z \in \mathcal{Z}} D_{z|y}(\hat{y})=1; \forall \hat{y}} \sum_{y \in \mathcal{Y}} \sum_{z \in \mathcal{Z}} P_{Y,Z}(y,z) \mathbb{E}_{P_{\hat{Y}|y,z}} \left[ \log D_{z|y}(\hat{Y}) \right] + H(Z|Y).$$

Recall that this conditional mutual information term can be used to capture *equalized odds*, which is another important fairness metric. We also note that the following theorem can be modified in a straightforward manner so that it can handle $I(Z; \hat{Y}|Y = 1)$, which can be used to capture *equal opportunity*.

We now prove the theorem.

*Proof.* Denote by $\boldsymbol{D}$ the collection of $D_{z|y}(\hat{y})$ for all possible values of $(z, \hat{y},$ and $y)$ and by $\boldsymbol{\nu}$ the collection of $\nu_{y,\hat{y}}$ for all values of $y$ and $\hat{y}$. We can construct the Lagrangian function as follows:

$$\mathcal{L}(\boldsymbol{D}, \boldsymbol{\nu}) = \sum_{y \in \mathcal{Y}} \sum_{z \in \mathcal{Z}} P_{Y,Z}(y, z) \mathbb{E}_{P_{\hat{Y}|y,z}} \left[ \log D_{z|y}(\hat{Y}) \right] + H(Z|Y) + \sum_{\hat{y} \in \mathcal{Y}} \sum_{y \in \mathcal{Y}} \nu_{y,\hat{y}} \left( 1 - \sum_{z \in \mathcal{Z}} D_{z|y}(\hat{y}) \right).$$

We use the following KKT conditions:

$$\frac{\partial \mathcal{L}(\boldsymbol{D}, \boldsymbol{\nu})}{\partial D_{z|y}(\hat{y})} = P_{Y,Z}(y, z) \frac{P_{\hat{Y}|y,z}(\hat{y})}{D^\star_{z|y}(\hat{y})} - \nu_{y,\hat{y}} = 0, \qquad \forall (\hat{y}, y, z) \in \mathcal{Y} \times \mathcal{Y} \times \mathcal{Z}$$

$$1 - \sum_{z \in \mathcal{Z}} D^\star_{z|y}(\hat{y}) = 0, \qquad \forall (\hat{y}, y) \in \mathcal{Y} \times \mathcal{Y}.$$

Solving the two equations, we obtain $\nu^\star_{y,\hat{y}} = P_{Y,\hat{Y}}(y, \hat{y})$ for all $(y, \hat{y}) \in \mathcal{Y} \times \mathcal{Y}$. Thus,

$$D^\star_{z|y}(\hat{y}) = \frac{P_{Z|y}(z) P_{\hat{Y}|y,z}(\hat{y})}{P_{\hat{Y}|y}(\hat{y})}, \forall y, \hat{y} \in \mathcal{Y} \times \mathcal{Y}.$$

Putting this to the above optimization,

$$\sum_{y \in \mathcal{Y}} \sum_{z \in \mathcal{Z}} P_{Y,Z}(y, z) \mathbb{E}_{P_{\hat{Y}|y,z}} \left[ \log \frac{P_{Z|y}(z) P_{\hat{Y}|y,z}(\hat{Y})}{P_{\hat{Y}|y}(\hat{Y})} \right] + H(Z|Y)$$

$$= \sum_{y \in \mathcal{Y}} \sum_{z \in \mathcal{Z}} P_{Y,Z}(y, z) \mathbb{E}_{P_{\hat{Y}|y,z}} \left[ \log \frac{P_{Z|y}(z) P_{\hat{Y}|y,z}(\hat{Y})}{P_{\hat{Y}|y}(\hat{Y})} \right] + \sum_{y \in \mathcal{Y}} \sum_{z \in \mathcal{Z}} P_{Y,Z}(y, z) \log \frac{1}{P_{Z|y}(z)}$$

$$= \sum_{y \in \mathcal{Y}} \sum_{z \in \mathcal{Z}} P_{Y,Z}(y, z) \mathbb{E}_{P_{\hat{Y}|y,z}} \left[ \log \frac{P_{\hat{Y}|y,z}(\hat{Y})}{P_{\hat{Y}|y}(\hat{Y})} \right]$$

$$= \sum_{y \in \mathcal{Y}} \sum_{z \in \mathcal{Z}} P_Y(y) P_{Z|y}(z) \mathbb{E}_{P_{\hat{Y}|y,z}} \left[ \log \frac{P_{\hat{Y}|y,z}(\hat{Y})}{P_{\hat{Y}|y}(\hat{Y})} \right]$$

$$= \sum_{y \in \mathcal{Y}} P_Y(y) \sum_{z \in \mathcal{Z}} P_{Z|y}(z) \mathbb{E}_{P_{\hat{Y}|y,z}} \left[ \log \frac{P_{\hat{Y}|y,z}(\hat{Y})}{P_{\hat{Y}|y}(\hat{Y})} \right]$$

$$= \sum_{y \in \mathcal{Y}} P_Y(y) \sum_{z \in \mathcal{Z}} P_{Z|y}(z) D_{\mathrm{KL}}(P_{\hat{Y}|y,z} \| P_{\hat{Y}|y})$$

$$\triangleq \sum_{y \in \mathcal{Y}} P_Y(y) \cdot \mathrm{JS}_{P_{Z|y}} \left( P_{\hat{Y}|z_1,y}, \ldots, P_{\hat{Y}|z_{|\mathcal{Z}|},y} \right)$$

$$= \sum_{y \in \mathcal{Y}} P_Y(y) I(Z; \hat{Y}|Y = y) = I(Z; \hat{Y}|Y).$$

The third last equality is due to the definition of the generalized JS divergence; the second last equality is due to its equivalence to the mutual information (Lin, 1991); and the last equality is due to the definition of conditional mutual information. $\square$

We now discuss how to actually compute the mutual information. We compute the following empirical version using the example $\{(x^{(i)}, z^{(i)}, y^{(i)})\}_{i=1}^m$.

$$\max_{D_{z|y}(\hat{y}) : \sum_{z \in \mathcal{Z}} D_{z|y}(\hat{y}) = 1; \forall \hat{y}} \sum_{y \in \mathcal{Y}} \sum_{z \in \mathcal{Z}} P_{Y,Z}(y, z) \sum_{i : (y^{(i)}, z^{(i)}) = (y, z)} \frac{1}{m_{y,z}} \log D_{z|y}(\hat{y}^{(i)}) + H(Z|Y).$$

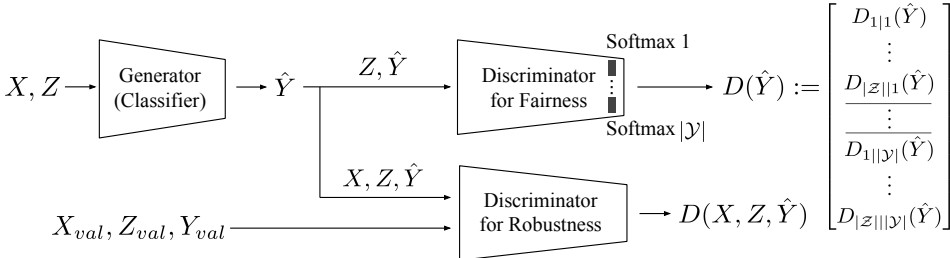

Figure 4: The architecture of FR-GAN for equalized odds.

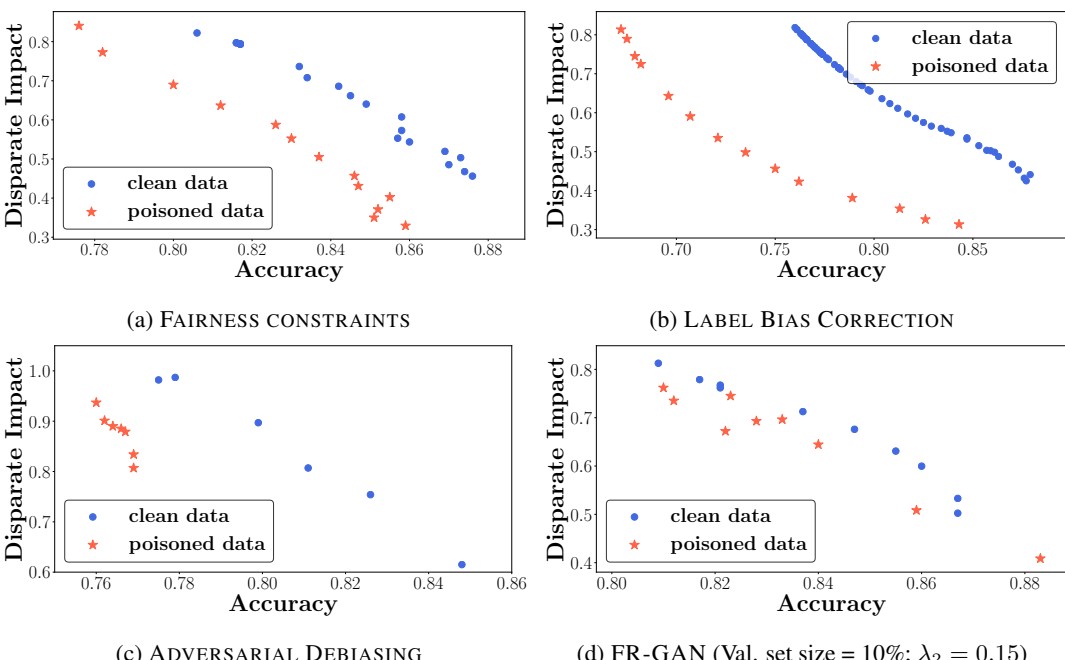

(a) FAIRNESS CONSTRAINTS

(b) LABEL BIAS CORRECTION

(c) ADVERSARIAL DEBIASING

(d) FR-GAN (Val. set size = 10%; $\lambda_2 = 0.15$)

Figure 5: Accuracy-fairness tradeoff curves for poisoned synthetic data with label flipping (7%).

Now for a sufficiently large value of $m$, $m_{y,z} \approx P_{Y,Z}(y,z)m$. Therefore, the above expression is approximated as:

$$\max_{D_{z|y}(\hat{y}): \sum_{z \in \mathcal{Z}} D_{z|y}(\hat{y})=1; \forall \hat{y}} \sum_{y \in \mathcal{Y}} \sum_{z \in \mathcal{Z}} \sum_{i:(y^{(i)},z^{(i)})=(y,z)} \frac{1}{m} \log D_{z|y}(\hat{y}^{(i)}) + H(Z|Y).$$

Hence, we can set $L_2$ (i.e., the loss w.r.t. the fairness discriminator) to the above expression. The rest of the objective function is the same. Figure 4 shows the resulting FR-GAN architecture.

### A.3 ADDITIONAL EXPERIMENTS

### A.3.1 SYNTHETIC DATA

We continue our experiments from Sections 3 and 5.1.

**Label flipping for data poisoning** Figure 5a to 5d show the experimental results when the data is poisoned via label flipping. We used the same synthetic data, as illustrated in Figure 1a, and flipped 7% of labels from the clean data to make a poisoned data. Figures 5a, 5b, and 5c show the accuracy-fairness tradeoff curves of FAIRNESS CONSTRAINTS, LABEL BIAS CORRECTION, and ADVERSARIAL DEBIASING, respectively. While the tradeoff curves in Figure 5a to 5c considerably shift to the left, Figure 5d shows that FR-GAN is noticeably robust against label flipping.

Table 3: Accuracy and fairness performances of the meta learning method by Ren et al. (2018) on the clean and poisoned synthetic datasets for different validation set sizes. We used the same $z$-flipping poisoning attack described in Section 3.

| Data | Validation set size | Disparate impact | Accuracy |
|---|---|---|---|
| Clean Data | 10% | 0.429 | 0.883 |
| Poisoned Data | 10% | 0.386 | 0.884 |
| | 5% | 0.401 | 0.886 |
| | 1% | 0.297 | 0.845 |
| | 0.5% | 0.269 | 0.829 |
| | 0.1% | 0.348 | 0.585 |

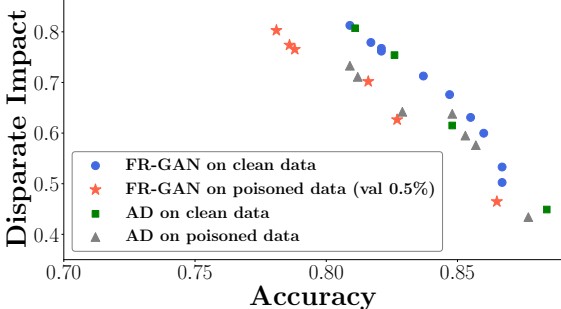

Figure 6: Comparison between ADVERSARIAL DEBIASING (AD) and FR-GAN with small validation set (0.5%) on synthetic data.

**Meta learning with different validation set sizes** Table 3 shows the accuracy and fairness results for the meta learning method by Ren et al. (2018) (ML) for different validation set sizes. We observe a drastic decrease of accuracy when the validation set size is 0.1% of the training data.

**Ablation study for fairness** Figure 6 shows the accuracy-fairness tradeoff curves for ADVERSARIAL DEBIASING and FR-GAN on the synthetic dataset. ADVERSARIAL DEBIASING tradeoff curves are from Figure 3b, and the FR-GAN curve comes from Figure 3f. We observe in Figure 6 that the accuracy-fairness tradeoff of FR-GAN using a small value of $\lambda_2$ on the poisoned data is similar to the results of fairness-only methods.

### A.3.2 REAL DATA

We continue our experiments from Section 5.2.

**Label flipping for data poisoning** Table 4 uses the same experimental setting as in Table 2, except that data poisoning is label flipping (not w.r.t. sensitive attributes). We observe similar trends as in Table 2. For the non-fairness methods, the fairness results are significantly worse than other methods. For the fairness-only methods, both fairness and accuracy tend to be high on the clean data, but at least one of them decreases significantly on the poisoned data. On the other hand, FR-GAN yields high fairness and accuracy on the clean data and is also the most resilient against poisoned data. The results emphasize our holistic technique that takes both fairness and accuracy into consideration.

### A.3.3 FR-GAN USING OTHER FAIRNESS MEASURES

As we showed in Appendix A.2, FR-GAN respects equalized odds and equal opportunity. Table 5 shows the experimental results on the synthetic and real datasets for equalized odds. We see that FR-GAN significantly improves equalized odds with reasonable accuracy. The results w.r.t. equal opportunity are similar and thus not shown here.

Table 4: Accuracy and fairness performances on real data for disparate impact where the data is poisoned using a label flipping attack (Paudice et al., 2018). Two types of methods are compared: (1) non-fairness methods: logistic regression (LR), the meta learning by Ren et al. (2018) (ML), and R-GAN (FR-GAN with $\lambda_1 = 0$); (2) fairness-only methods: FAIRNESS CONSTRAINTS (Zafar et al., 2017) (FC), LABEL BIAS CORRECTION (Jiang & Nachum, 2019) (LBC), and ADVERSARIAL DEBIASING (Zhang et al., 2018) (AD). Also "+ LD" denotes loss defense (Koh et al., 2018) (a data sanitization technique that is applied prior to model training). For FR-GAN, the validation set is 10% of $\mathcal{D}_{tr}$. For each fairness or accuracy result of the poisoned data, we make a comparison with the clean data result and show the percentage increase or decrease.

| Dataset | Type | Method | Clean data | | Poisoned data | |
|---|---|---|---|---|---|---|
| | | | D. impact | Acc. | D. impact | Acc. |
| COMPAS | Non-fair | LR | 0.465 | 0.674 | 0.856 (84.1% ↑) | 0.487 (27.7% ↓) |
| | | ML | 0.493 | 0.680 | 0.608 (23.3% ↑) | 0.629 (7.50% ↓) |
| | | R-GAN | 0.574 | 0.683 | 0.919 (60.1% ↑) | 0.621 (9.08% ↓) |
| | Fair-only | FC | 0.777 | 0.682 | 0.931 (19.8% ↑) | 0.490 (28.2% ↓) |
| | | LBC | 0.866 | 0.671 | 0.945 (9.12% ↑) | 0.488 (27.3% ↓) |
| | | AD | 0.846 | 0.680 | 0.941 (11.2% ↑) | 0.615 (9.56% ↓) |
| | | FC + LD | 0.777 | 0.682 | 0.820 (5.53% ↑) | 0.544 (20.2% ↓) |
| | | LBC + LD | 0.866 | 0.671 | 0.928 (7.16% ↑) | 0.550 (18.0% ↓) |
| | | AD + LD | 0.846 | 0.680 | 0.822 (2.84% ↓) | 0.614 (9.71% ↓) |
| | Both | FR-GAN | 0.936 | 0.667 | 0.964 (2.99% ↑) | 0.621 (6.90% ↓) |
| Adult | Non-fair | LR | 0.328 | 0.847 | 0.316 (3.66% ↓) | 0.824 (2.72% ↓) |
| | | ML | 0.346 | 0.844 | 0.413 (19.4% ↑) | 0.826 (2.13% ↓) |
| | | R-GAN | 0.335 | 0.847 | 0.342 (2.09% ↑) | 0.833 (2.13% ↓) |
| | Fair-only | FC | 0.825 | 0.826 | 0.814 (1.33% ↓) | 0.802 (2.91% ↓) |
| | | LBC | 0.825 | 0.825 | 0.734 (11.0% ↓) | 0.819 (0.73% ↓) |
| | | AD | 0.850 | 0.767 | 0.652 (23.3% ↓) | 0.749 (2.35% ↓) |
| | | FC + LD | 0.825 | 0.826 | 0.762 (7.64% ↓) | 0.831 (0.61% ↑) |
| | | LBC + LD | 0.825 | 0.825 | 0.720 (12.7% ↓) | 0.827 (0.24% ↑) |
| | | AD + LD | 0.850 | 0.767 | 0.761 (10.5% ↓) | 0.803 (4.69% ↑) |
| | Both | FR-GAN | 0.870 | 0.819 | 0.859 (1.26% ↓) | 0.789 (3.66% ↓) |

Table 5: Accuracy and fairness performances on synthetic and real datasets w.r.t. equalized odds. Two algorithms are compared: (1) logistic regression (LR, non-fairness method) and (2) FR-GAN.

| Dataset | Method | Equalized odds | | Accuracy |
|---|---|---|---|---|
| | | $Y = 0$ | $Y = 1$ | |
| Synthetic Data | LR | 0.351 | 0.804 | 0.885 |
| | FR-GAN | 0.888 | 0.936 | 0.865 |
| Compas | LR | 0.427 | 0.557 | 0.674 |
| | FR-GAN | 0.718 | 0.959 | 0.628 |
| AdultCensus | LR | 0.286 | 0.909 | 0.848 |
| | FR-GAN | 0.503 | 0.917 | 0.842 |

## A.4 TRAINING METHODOLOGY

We implement all algorithms in PyTorch (Paszke et al., 2017), and all experiments are performed on a server with Intel i7-6850 CPUs.

The generator $G$ is a neural network with zero or one hidden layer. The discriminator $D^f$ is a single layer neural network, and the discriminator $D^r$ is a neural network with one hidden layer. We used 8 or 16 nodes in the hidden layers. We set an Adam optimizer (Kingma & Ba, 2014) for the generator,

Table 6: Confusion matrix on poisoned AdultCensus dataset w.r.t. disparate impact. Two algorithms are compared: (1) AD and (2) FR-GAN. For a fair comparison, we use data sanitization before running AD.

| | | Female | | Male | |
|---|---|---|---|---|---|
| Method | | Prediction = 0 | Prediction = 1 | Prediction = 0 | Prediction = 1 |
| AD + LD | True = 0 | 2,502 | 566 | 4,330 | 583 |
| | True = 1 | 191 | 194 | 1,551 | 714 |
| FR-GAN | True = 0 | 2,765 | 303 | 4,611 | 302 |
| | True = 1 | 95 | 290 | 1,144 | 1,121 |

Table 7: Confusion matrix on poisoned AdultCensus dataset w.r.t. equalized odds. Two algorithms are compared: (1) AD and (2) FR-GAN. For a fair comparison, we use data sanitization before running AD.

| | | Female | | Male | |
|---|---|---|---|---|---|
| Method | | Prediction = 0 | Prediction = 1 | Prediction = 0 | Prediction = 1 |
| AD + LD | True = 0 | 2,935 | 133 | 4,629 | 284 |
| | True = 1 | 249 | 136 | 1,590 | 675 |
| FR-GAN | True = 0 | 2,859 | 209 | 4,480 | 443 |
| | True = 1 | 116 | 269 | 960 | 1,305 |

and a stochastic gradient descent (SGD) optimizer for each discriminator. We empirically observe that one can stabilize the training procedure by freezing the parameters of the fairness discriminator $D^f$ for the initial phase of training. Thus, we choose to freeze the parameters of the fairness discriminator $D^f$ for the first few epochs until the generator achieves a certain accuracy. We use the generator/discriminator update ratio of 1:1 for the first 300 epochs and use the update ratio of 1:3 (or 1:5) for the rest of training.

Also, we use the following details for choosing the values of $\lambda_1$ and $\lambda_2$. For clean data, we set $\lambda_2$ as a small value (e.g., 0.01) and vary $\lambda_1$ from 0 to 0.95. For poisoned data, we set $\lambda_2$ as 0.2, 0.3, or 0.4, and vary $\lambda_1$ from 0 to $0.95 - \lambda_2$. Given the values of $\lambda_1$ and $\lambda_2$, we also normalize $L_1$ (the generator loss) by multiplying it with $(1 - \lambda_1 - \lambda_2)$.

## A.5 ADDITIONAL EXPERIMENTS ON ADULT DATASET

We compare FR-GAN and AD using the AdultCensus dataset and the same poisoning in Section 3. For a fair comparison, we use loss defense (LD) as data sanitization prior to running AD. We generate confusion matrices both for disparate impact (DI) and equalized odds (EO). Overall, FR-GAN performs better than AD because of its robustness discriminator. The robustness discriminator successfully ignores the poisoned distribution in the training data, and FR-GAN's TPR is higher than AD with sanitization. Table 6 shows confusion matrices as results of disparate-impact-focused experiment. The results are reported when AD achieves (Acc, DI) = (0.728, 0.821), and FR-GAN achieves (Acc, DI) = (0.827, 0.866). In addition, Table 7 represents confusion matrices as results of equalized-odds-focused experiment. The results are reported when AD achieves (Acc, EO when $y = 1$, EO when $y = 0$) = (0.788, 0.844, 0.750), and FR-GAN achieves (Acc, EO when $y = 1$, EO when $y = 0$) = (0.838, 0.824, 0.773).

