# OpenReview forum: "FR-GAN: Fair and Robust Training"
_ICLR.cc/2020/Conference — Reject_

### Official Review · AnonReviewer3 · 2019-10-22
**Official Blind Review #3**

**Rating:** 3

**Review:**

This paper introduces a new method for training a classifier that simultaneously optimizes for a fairness criterion and robustness to data poisoning. The method is shown to increase measures of fairness and reduce inaccuracy on poisoned data relative to classifiers that only consider accuracy or fairness. Extensive results are shown for both synthetic and real benchmark data sets.

I would lean to reject for the following reasons: 1) the problem is not well-motivated. I would like a more clear example of some problem with sensitive attributes in which the data is publicly available and the providers of the data are motivated to falsify it. 2) the contribution is very simple and the individual pieces do not seem to be significant contributions. In particular, the use of GANs for fairness is previously done, and the use of the GAN for robustness here seems too simple to be broadly useful 3) the results are less convincing than they might otherwise be because none of the competing methods tested make use of a clean validation set 4) the paper is somewhat unpolished. I find the results difficult to read, although the arrows are helpful, and it is not clear to me whether these results are on a test set or the training set.

Lack of convincing tests for robustness: It is disappointing that FR-GAN does not offer any promises to be robust in general. Despite access to a clean validation set, the classifier is trained only to ignore the type of data poisoning that exists in the training set. If the test set were out of distribution in a different way relative to the training set, I see no reason to believe FR-GAN would protect against this. Furthermore, because it is not stated that these are test set results, I am not certain that they are not training set results, in which case some performance may be due to overfitting.

Minor notes:

It would be nice for comparison if the charts had the same axes throughout.

What are the numbers of nodes used in the hidden layers?

**Experience Assessment:**

I do not know much about this area.

**Review Assessment: Checking Correctness Of Derivations And Theory:**

I did not assess the derivations or theory.

**Review Assessment: Checking Correctness Of Experiments:**

I assessed the sensibility of the experiments.

**Review Assessment: Thoroughness In Paper Reading:**

I read the paper at least twice and used my best judgement in assessing the paper.

---

> ### Author Response · Authors · 2019-11-13
> **Response to Reviewer 3**
>
> Thanks for the insightful comments.
>
> Q3-1. (1) motivation
>
> A3-1.
> We believe that given that real data would increasingly become both biased and poisoned (this is what we expect in the big data era - see the next paragraph for details), our main contribution of providing an integrated solution for fair and robust training is likely to become important in the near future.
>
> We contend that supporting robust training is just as critical as fair training. Dataset searching is becoming mainstream as demonstrated by Google Dataset Search (Goods) [1] and its public version for searching scientific datasets [2]. While data lakes within companies may consist of refined datasets, datasets in the public are easy to poison. In our experiments, we could easily poison the Adult and COMPAS datasets using simple label flipping techniques. And anyone can poison public datasets using attacks in the literature and share them. We thus believe that it is essential to address both bias and poisoning as a preventive measure. We reflected these points in our revision (Section 1, highlighted in blue).
>
> [1] Goods: Organizing Google's Datasets, ACM SIGMOD 2017.
> [2] Google Dataset Search: Building a search engine for datasets in an open Web ecosystem, WWW 2019.
>
>
> Q3-2. (2) contributions
>
> A3-2.
> We agree that the fairness part of FR-GAN is similar to Adversarial Debiasing (AD) [3]. However, we strengthen the theoretical results of AD using information theory, and this motivates us to propose a novel robust training discriminator. We agree that the key insight of using adversarial training to minimize mutual information between the prediction and sensitive attribute is the same for FR-GAN and AD. Although, the end algorithms of FR-GAN and AD are also similar, our paper plays a role in providing systematic methodology not only for various fairness metrics, but also for the design of robust training. As far as we know, no other validation-set-based approach (including Ren et. al. 2018) leverages the idea of adversarial training. We reflected these points in our revision (Section 2, highlighted in blue).
>
> [3] Mitigating Unwanted Biases with Adversarial Learning, AAAI 2018.
>
>
> Q3-3. (3) competing methods
>
> A3-3.
> As Reviewer 2 mentioned, FR-GAN is one of the first works to address both fairness and robustness and that there are few baselines to compare with. Hence, we employed one reasonable baseline, which first sanitizes the poisoned data using a well-known sanitization technique and then performs a fair training (see Tables 1 and 2, rows with +LD). We clarified these points in our revision (Section 5.1, highlighted in blue).
>
>
> Q3-4. (4) writeup
>
> A3-4.
> All of our results are on a separate test set. We clarified this in our revision (Section 5, highlighted in blue) to avoid any confusion.
>
>
> Q3-5. Lack of convincing tests for robustness
>
> A3-5.
> We agree that generalizing to all poisoning attacks is important. If we know all possible attacks, we can construct a training set containing these attacks. In the more challenging case where we do not know which attacks even exist, there seems to be a fundamental limitation in protecting against the attacks. Generalization is a universal problem in machine learning where a model trained on one dataset is not guaranteed to perform well in another dataset with a different distribution. Although the generalization is a critical issue to address, we think it is beyond the scope of the current work. We reflected these points in our revision (Section 6, highlighted in blue).
>
>
> Q3-6. Minor notes
>
> A3-6.
> We will display the figures with the same axes throughout in our revision. We added the information about the number of nodes in the hidden layers in our revision (Appendix A.4, highlighted in blue).

---

### Official Review · AnonReviewer2 · 2019-10-23
**Official Blind Review #2**

**Rating:** 3

**Review:**

This paper combines adversarial fair training with adversarial robust training. The basic idea is that a classifier is combined with two adversaries: one tries to predict the sensitive attribute $Z$ from the output of the classifier (essentially the approach by Edwards&Storkey 2015) and the other adversary tries to recognize if a label was predicted or is from a clean hold-out dataset. The latter is intended to harden the classifier against data-poisoning of the training set.

---

The paper is clearly written and technically sound.

The fairness aspect of the proposed method is fairly standard and not very novel (going back to 2015). The robustness aspect is an interesting addition and seems novel, but I'm not sure if it's enough to get the paper accepted. If I understand it correctly, FR-GAN without the "R" part should just be equivalent to Adversarial Debiasing (Zhang et al., 2018). And if there is no data-poisoning, then the "R" part doesn't have any effect.

The fact that this is the first fairness-related method that additionally deals with robustness, makes it also difficult to judge the performance of the method. I would wish for a more appropriate baseline; one that makes use of the clean validation set somehow.

There might be synergistic effects where the robustness aspects helps the fairness aspect but this comes at the cost of needing a clean validation set (and it only matters with poisoned data).

What is also missing is a motivating real-world example. When would you encounter flipped labels in the training set, but also have access to a clean validation set?

Minor comments:

- I did not understand what was meant by the phrase "so that the model accuracy is reduced the most." in the first paragraph of section 3

**Experience Assessment:**

I have published one or two papers in this area.

**Review Assessment: Checking Correctness Of Derivations And Theory:**

I assessed the sensibility of the derivations and theory.

**Review Assessment: Checking Correctness Of Experiments:**

I assessed the sensibility of the experiments.

**Review Assessment: Thoroughness In Paper Reading:**

I read the paper at least twice and used my best judgement in assessing the paper.

---

> ### Author Response · Authors · 2019-11-13
> **Response to Reviewer 2**
>
> Thanks for the insightful comments.
>
> Q2-1. Novelty and Motivation
>
> A2-1.
> We agree that the fairness part of FR-GAN is similar to Adversarial Debiasing (AD). However, we believe that given that real data would increasingly become both biased and poisoned (this is what we expect in the big data era - see the next paragraph for details), our main contribution of providing an integrated solution for fair and robust training is likely to become important in the near future.
>
> We contend that supporting robust training is just as critical as fair training. Dataset searching is becoming mainstream as demonstrated by Google Dataset Search (Goods) [1] and its public version for searching scientific datasets [2]. While data lakes within companies may consist of refined datasets, datasets in the public are easy to poison. In our experiments, we could easily poison the Adult and COMPAS datasets using simple label flipping techniques. And anyone can poison public datasets using attacks in the literature and share them. We thus believe that it is essential to address both bias and poisoning as a preventive measure.
>
> Regarding technical contributions, we strengthen the theoretical results of AD using information theory, and this motivates us to propose a novel robust training discriminator. We agree that the key insight of using adversarial training to minimize mutual information between the prediction and sensitive attribute is the same for FR-GAN and AD. Although, the end algorithms of FR-GAN and AD are also similar, our paper plays a role in providing systematic methodology not only for various fairness metrics, but also for the design of robust training. As far as we know, no other validation-set-based approach (including Ren et. al. 2018) leverages the idea of adversarial training.
>
> We reflected all of the points above in our revision (Sections 1 and 2, highlighted in blue).
>
> [1] Goods: Organizing Google's Datasets, ACM SIGMOD 2017.
> [2] Google Dataset Search: Building a search engine for datasets in an open Web ecosystem, WWW 2019.
>
>
> Q2-2. Baseline
>
> A2-2.
> As Reviewer 2 mentioned, FR-GAN is one of the first works to address both fairness and robustness and that there are few baselines to compare with. Hence, we employed one reasonable baseline, which first sanitizes the poisoned data using a well-known sanitization technique and then performs a fair training (see Tables 1 and 2, rows with +LD). We clarified these points in our revision (Section 5.1, highlighted in blue).
>
>
> Q2-3. Motivating real-world example
>
> A2-3.
> Unfortunately, there is no such public poisoned dataset in the context of fairness training. We believe this is because: 1) the fair-&-robust training of our focus is a new topic; and 2) no adversary would explicitly claim that s/he poisoned a dataset.
>
> We agree with Reviewer 1 that FR-GAN can be generalized to a setting where labels are subjective instead of being clean or poisoned. Here the dataset without the undesirable human biases becomes the "clean" validation set. Inspired by this idea, we would like to propose a new method for constructing a real dataset. Suppose that we are indeed performing loan decisions where there is a high chance of human bias (i.e., poisoning) due to the high workload. We can construct a clean validation set by selecting a few loans, assigning more evaluators, and taking a majority vote of the evaluations to minimize the bias. While employing more evaluators can be expensive, constructing a small validation set is sufficient for robust training, which makes this method practical. We reflected these points in our revision (Section 6, highlighted in blue).
>
>
> Q2-4. Minor comment
>
> A2-4.
> We rephrased "so that the model accuracy is reduced the most" in Section 3 to "To generate poisoned data, we measure accuracy degradation for a flipping of each Z. We then choose the top-10% of the Z values with the highest degradation and flip them." (Section 3, highlighted in blue). Thanks for pointing this out.

---

### Official Review · AnonReviewer1 · 2019-10-29
**Official Blind Review #1**

**Rating:** 3

**Review:**

This paper proposes to use a GAN style approach for training a classifier that is robust to data poisoning and can achieve a pre-specified notion of group fairness.

The contribution of this paper is incremental in the context of prior Adversarial Debasing (AD) approach using essentially same GAN for group fairness and prior work presenting ideas of utilizing clean validation data to defend against data poisoning. This paper is proposing to add an additional discriminator to the AD approach that distinguishes training data and clean validation data. If the training data is poisoned, such distinguishment may be possible and maximizing loss of this discriminator can help to robustify against poisoned samples.

Experimental results are insufficient to argue improvement over the AD. There are no AD results in the equalized odds Adult experiment in the supplement. I recommend more detailed comparison against the AD method (including results showing confusion matrices). Also note that AD, as presented in the original paper, is optimizing for demographic parity, but can also be adjusted to other group fairness metrics. Finally, in the context of Adult dataset, it is important to also report performance metrics such as balanced TPR since the labels are quite imbalanced.

Are there any real data examples where poisoning does not need to be introduced artificially and the proposed method helps to improve the fairness properties? I think an interesting direction could be to consider data where labels are subjective. For example, a dataset on loan decisions can be naturally "poisoned" with human biases, i.e. information that someone did not receive the loan may be due to an error or bias of a human in charge of the decision making.

Lastly I think that the discussion of the prior related work on fairness is incomplete. This paper exclusively covers group fairness, which indeed has been shown to have some disadvantages. For example, prior work [1] has shown that some group fairness notions can not be satisfied simultaneously in certain cases. In this regard it is also important to report multiple group fairness metrics simultaneously in the experiments. The other fairness definition, that has not been mentioned in this paper, is individual fairness [2]. It has legal and intuitive interpretations. Multiple recent papers have explored the direction of individual fairness [3,4,5,6].

[1] Kleinberg, J., Mullainathan, S., & Raghavan, M. (2016). Inherent trade-offs in the fair determination of risk scores.
[2] Dwork, C., Hardt, M., Pitassi, T., Reingold, O., & Zemel, R. (2012, January). Fairness through awareness.
[3] Garg, S., Perot, V., Limtiaco, N., Taly, A., Chi, E. H., & Beutel, A. (2019, January). Counterfactual fairness in text classification through robustness.
[4] Yurochkin, M., Bower, A., & Sun, Y. (2019). Learning fair predictors with Sensitive Subspace Robustness.
[5] Kearns, M., Roth, A., & Sharifi-Malvajerdi, S. (2019). Average Individual Fairness: Algorithms, Generalization and Experiments.
[6] Jung, C., Kearns, M., Neel, S., Roth, A., Stapleton, L., & Wu, Z. S. (2019). Eliciting and Enforcing Subjective Individual Fairness.

**Experience Assessment:**

I have published one or two papers in this area.

**Review Assessment: Checking Correctness Of Derivations And Theory:**

I did not assess the derivations or theory.

**Review Assessment: Checking Correctness Of Experiments:**

I assessed the sensibility of the experiments.

**Review Assessment: Thoroughness In Paper Reading:**

I read the paper at least twice and used my best judgement in assessing the paper.

---

> ### Author Response · Authors · 2019-11-13
> **Response to Reviewer 1**
>
> Thanks for the insightful comments.
>
> Q1-1. Motivation and Contributions
>
> A1-1.
> We agree that the fairness part of FR-GAN is similar to Adversarial Debiasing (AD). However, we believe that given that real data would increasingly become both biased and poisoned (this is what we expect in the big data era - see the next paragraph for details), our main contribution of providing an integrated solution for fair and robust training is likely to become important in the near future.
>
> We contend that supporting robust training is just as critical as fair training. Dataset searching is becoming mainstream as demonstrated by Google Dataset Search (Goods) [1] and its public version for searching scientific datasets [2]. While data lakes within companies  may consist of refined datasets, datasets in the public are easy to poison. In our experiments, we could easily poison the Adult and COMPAS datasets using simple label flipping techniques. And anyone can poison public datasets using attacks in the literature and share them. We thus believe that it is essential to address both bias and poisoning as a preventive measure.
>
> Regarding technical contributions, we strengthen the theoretical results of AD using information theory, and this motivates us to propose a novel robust training discriminator. We agree that the key insight of using adversarial training to minimize mutual information between the prediction and sensitive attribute is the same for FR-GAN and AD. Although, the end algorithms of FR-GAN and AD are also similar, our paper plays a role in providing systematic methodology not only for various fairness metrics, but also for the design of robust training. As far as we know, no other validation-set-based approach (including Ren et.al. 2018) leverages the idea of adversarial training.
>
> We reflected all of the points above in our revision (Sections 1 and 2, highlighted in blue).
>
> [1] Goods: Organizing Google's Datasets, ACM SIGMOD 2017.
> [2] Google Dataset Search: Building a search engine for datasets in an open Web ecosystem, WWW 2019.
>
>
> Q1-2. Experiments
>
> A1-2.
> As per your great comment, we now performed more detailed comparisons with AD. In particular, we generated confusion matrices both for disparate impact and equalized odds. As a result, we find FR-GAN performs better than AD under such measures, perhaps due to our robustness discriminator component (please see Appendix A.5 in the revised version of our paper, highlighted in blue). The robustness discriminator successfully ignores the poisoned distribution in the training data, and FR-GAN's TPR is higher than AD with sanitization. We may further refine our results later.
>
>
> Q1-3. Real-world examples
>
> A1-3.
> Unfortunately, there is no such public poisoned dataset in the context of fairness training. We believe this is because: 1) the fair-&-robust training of our focus is a new topic; and 2) no adversary would explicitly claim that s/he poisoned a dataset.
>
> We fully agree that FR-GAN can be generalized to a setting where labels are subjective instead of being clean or poisoned. Here the dataset without the undesirable human biases becomes the "clean" validation set. Inspired by this idea, we would like to propose a new method for constructing a real dataset. Suppose that we are indeed performing loan decisions where there is a high chance of human bias (i.e., poisoning) due to the high workload. We can construct a clean validation set by selecting a few loans, assigning more evaluators, and taking a majority vote of the evaluations to minimize the bias. While employing more evaluators can be expensive, constructing a small validation set is sufficient for robust training, which makes this method practical.
>
> In general, given two datasets with different distributions where one of them is desirable, FR-GAN can train robustly against the other distribution.
>
> We reflected all the points above in our revision (Section 6, highlighted in blue).
>
>
> Q1-4. Related work
>
> A1-4.
> We thank the reviewers for introducing papers on individual fairness. We were actually aware of this literature, but since FR-GAN is solving a new problem, our first step was to use prominent measures for group fairness, just like the AD paper. We mentioned all the individual fairness measures in our revision (Section 2, highlighted in blue) and will investigate them in a future work.

---

### Decision · Program_Chairs · 2019-12-19

**Decision:**

Reject

**Comment:**

This manuscript proposes an approach for fair and robust training of predictive modeling -- both of which are implemented using adversarial methods, i.e., an adversarial loss for fairness and an adversarial loss for robustness. The resulting model is evaluated empirically and shown to improve fairness and robustness performance.

The reviewers and AC agree that the problem studied is timely and interesting, as there is limited work on joint fairness and robustness. However, the reviewers were unconvinced about the novelty and clarity of the conceptual and empirical results. In reviews and discussion, the reviewers also noted insufficient motivation for the approach.